# Quality Assurance for Performing Arts Education: A Multi-Dimensional Analysis Approach

**Qingyun Li [1], Zihao (Michael) Li [2,3,*], Jie Han [4] and Huimin Ma [1]**

1   School of Management and Engineering, Xuzhou University of Technology, Xuzhou 221006, China; qingyun.li@gmail.com (Q.L.); mahuimin@xzit.edu.cn (H.M.)
2   The Hong Kong Academy for Performing Arts, Hong Kong SAR, China
3   Faculty of Education, University of Macau, Avenida da Universidade, Taipa, Macau
4   School of Science and Technology, Hong Kong Metropolitan University, Hong Kong SAR, China; chan@hkmu.edu.hk
*   Correspondence: michaelli@hkapa.edu

**Abstract:** Senior management in tertiary institutions desires an efficient system that could help them assess and evaluate learning outcomes so that effective policies can be implemented to enhance teaching and learning. This gets intensified as broader issues arise and higher expectations are put on tertiary education—build a creative workforce and adapt to new technologies to analyze the large volume of teaching and learning data. Government and higher education policymakers have to rapidly adjust relevant policies to surmount the challenges from the pandemic and also to keep up with technological advancement. This demands a novel and efficient way for policymakers and senior management to see and gain insights from a large volume of data (e.g., student course and teacher evaluation). In this study, the researchers present such a system through various examples. The findings generated from this study contribute to the scholarship, and they provide a solution to senior management in tertiary institutions wanting to implement effective policies efficiently. The use of online analytical processing, virtual campus, online, and machine learning in education is growing. However, the use of these technology-enhanced approaches is rare in performing arts education. There has been no in-depth study, especially on technology-enhanced learning that leads to the improvement of teaching. This study utilizes a multi-dimensional analysis approach on the course student evaluation, a key aspect of the teaching and learning quality assurance for higher education. A novel analytical framework is developed and implemented at a leading performing arts university in Asia. It analyzes the course evaluation data of all courses (669 courses and 2664 responses) in the academic year 2018/2019 to make evidence-based recommendations. Such a framework provides an easy and effective visualization for senior management to identify courses that need closer scrutiny to ascertain whether and what areas of course enhancement measures are warranted.

**Keywords:** quality assurance; higher education; multi-dimensional analysis; analytical framework

## 1. Introduction

The student course evaluation survey exercises serve as one of the key student feedback channels for quality assurance and continuous enhancement in teaching and learning at the institutional level. Research shows that students' engagement in learning is a critical indicator of learning productivity, satisfaction, and academic success. Ref. [1] suggest that student engagement is a key indicator of the quality of their learning experience in school. Student engagement is referred to as the "effort to study a subject, practice, obtain feedback, analysis, and solve problems" by [2]. The work by [3] classifies student engagement into behavioral, emotional, cognitive, and agentic levels. In other words, student engagement is closely associated with their class participation, contribution, and self-ownership in learning.

The previous empirical analysis at a leading performing arts university in Asia shows that students' overall ratings of Outcome Achievement, Teacher Rating, and Student Effort attribute positively to Course Satisfaction. Meanwhile, Course Design, Teaching Practice, and Student Effort contribute positively to students' ratings on Outcome Achievement, Teacher Rating, and Course Satisfaction. Course Design has a significant impact on Course Satisfaction, whereas Teaching Practice leads to drastic variations in Teachers' Rating, especially in the performing arts education. These findings are consistent with the extant literature that fosters students' active engagement in learning, which is central to student satisfaction, learning productivity, and academic success.

Built on these insights, this study focuses on the following areas: (1) applies the data warehouse multi-dimensional analysis approach to analyze the course student evaluation data, a key aspect of the quality assurance for higher education; (2) applies a newly developed analytical framework to categorize courses into various domains (Student Effort, Outcome Achievement, Course Rating, and Teacher Rating) to address the domain-specific course issues for the program leaders and policymakers at the senior management level. In particular, this diagnostic tool provides a multi-dimensional lens to classify courses into different domains, which comprise metaphors and targeted enhancement strategies for addressing domain-specific issues.

## 2. Literature Review

Student evaluations of courses and teaching, as a key aspect of the teaching and learning quality assurance at tertiary institutions, have become increasingly important and is recognized as one of the ways in which teacher effectiveness can be identified and improved. Higher education institutions use the evaluation not only to assist in accreditation procedures and to provide measures for accountability but also to garner data regarding teaching quality, effectiveness, and enhancement.

A student satisfaction study conducted at a Romanian university by [4] presents important aspects in three dimensions (the teaching and learning activities, the institutional material base, and the support services offered by the institution) of the student satisfaction survey, and also shows that the evaluation is a suitable measure of effectiveness [5]. Various scaled studies were investigated to measure academics' performance based on student evaluation and indicated that the evaluation plays a significant role in academics' promotion or dismissal [6–8].

Furthermore, the works by [9,10] explore relevant legal issues on the dismissal of the unproductive lecturers that do not meet the required evaluation expectations during times of restructuring in the context of the US or Australian education systems.

To elicit the course-related expectations of university students about elective courses in music, descriptive research and data are collected by a questionnaire at Ankara University by [11] on 552 students; it investigates: (1) Reasons behind choosing the music course; (2) Expectation of the objectives of the music course; (3) Expectation of the content of the music course; (4) Expectation of the teaching process of the music course; and (5) Expectation of the evaluation process of the music course.

Interpretative Phenomenological Analysis (IPA) is used in music education research by [12] to investigate the personal meaning and lived experience in the music classroom. The study examines musical identity, detail of curriculum, pedagogy, and technology. This study explores how the analysis system works in the relevant research for music educators.

The author of [13] explores the potential for using a synchronous online piano teaching internship as a service-learning project for graduate pedagogy interns. The work suggests that educators shift the teaching focus from teacher-centered to student-centered (from single-way interaction to bilateral interaction). Thus, feedback from students becomes more important. The study shows evidence of an effective learning and engaging classroom. It is beneficial to hear from learners to manifest learning outcomes through improved teaching.

The work by [14] investigates the impact the pandemic has had on Ethno World, JM-International's programme for folk, world, and traditional music. The research aims

to find out how artistic mentors have responded to the teaching and learning shift from face-to-face to an online environment. A range of questions are used, including: How do artistic mentors perceive online teaching and learning practices during COVID-19? How has the shift to an online music teaching and learning environment impacted their understanding of the teaching and learning principles of Ethno? The study has revealed both pros and cons of online learning.

The works of [3,15] reveal that students are motivated to engage in learning when those learning activities and social conditions satisfy their basic needs for competence (feeling competent), autonomy (feeling in control), and relatedness (establishing emotional bonds with others). In teaching and learning contexts, students experience autonomy to the extent to which their learning activities foster a sense of choice, psychological freedom, and internal locus of control. As such, student engagement is affected by course design (curriculum development), learning environment (social climate), and social agents (teachers and peers), according to the works by [1,16]. Studies suggest that supportive teaching practices are positively associated with student engagement, including: (a) target high-order cognitive skills; (b) incorporate active learning activities; (c) involve collaborative investigation; and (d) incorporate social learning activities, such as observation, guided inquiry, and interaction with peers, experts, and teachers [2,17–19]. Furthermore, from the program design and management point of view, a combination of effective course designs and engaging teaching is key to student satisfaction, which leads to the higher achievement of learning goals.

Several newer evaluation models of teachings and a pathway to more accurate assessments are proposed by [20] to standardize the evaluation protocols for the performing arts teacher preparation programs. However, how to analyze the massive amount of data and how to present the aggregate data to others in an efficient way are extremely difficult for the evaluation of performing arts teachers.

With the rapid development of information technology, researchers try to use various online analytical processing technology and visualization methodologies in tertiary education, especially in the teaching and learning-related fields. Researchers [21] take a systematic review of learning analytic visualizations and conclude that while there is considerable work in the field, there "is a lack of studies that both employ sophisticated visualizations and engage deeply with educational theories" (p. 129). At the program level, scholars [22] present how learning analytics can inform the curriculum review through the analysis of data such as students' grades and subject satisfaction scores to identify areas for enhancement and improvement. The work by [23] presents the use of a data warehouse (DW) to analyze the behavior of the users (students) of the e-learning platform to make decisions with respect to their assessment. The DW is used to process and analyze data, which were generated by the students whilst navigating the e-learning platform. Ref. [24] uses the Online Analytical Processing (OLAP) technique in order to generate valuable reports, which are then used to improve the e-learning platform and help in learning evaluation.

Referring to the current online analytical processing technology and growing demands from the program leaders and policymakers at the senior management level, an efficient solution is needed. In this study, the researchers present a multi-dimensional analysis methodology and a novel analytical framework to categorize courses into various domains to address the domain-specific course issues. This framework results in an easy and effective visualization for relevant administrators and policymakers to quickly identify issues, adjust policies, and take necessary actions to improve and enhance teaching.

## 3. Data and Methodology

### 3.1. Source Data

This study focuses on a leading performing arts university in Asia. It is committed to maintaining the highest program and teaching standards. Student survey exercises are conducted regularly at the institutional level to gather feedback on faculty performance.

The centralized Student Feedback Questionnaire ("SFQ") survey is one of the means of the institutional evaluation system for quality assurance and continuous teaching enhancement.

The questionnaire covers questions in four dimensions—Student Effort, Outcomes Achievement, Course Rating, and Teacher Rating. The aspect-question mapping is shown in Table 1.

**Table 1.** The key aspect and mapping question in SFQ.

| Aspect | Question |
|---|---|
| Student Effort | Overall, I would rate my effort throughout the course as: |
| Outcomes Achievement | Overall, I achieved the learning outcomes of the course: |
| Course Rating | Overall, I would rate the course as: |
| Teacher Rating | Overall, I would rate the teacher as: |

The detailed questionnaire is listed in Appendix A. As a standard practice, two rounds of the SFQ survey were conducted for AY 2018/19, one to cover courses delivered in the Spring Semester and the other to cover courses delivered in the Fall Semester (summary in Table 2).

**Table 2.** Response rate and Mean Scores of students' responses * of SFQ 2018/19.

| | Semester 1 | Semester 2 |
|---|---|---|
| Number of Courses | 283 | 386 |
| Number of Eligible Respondent Count | 5545 | 6540 |
| Number of Actual Respondent Count | 1477 | 1187 |
| Response Rate | 27.1% | 18.2% |
| Mean Scores of students' responses * | | |
| Overall, I would rate my effort through the course as: | 4.95 | 4.90 |
| Overall, by the end of the course I had acquired the expected knowledge/skills: | 4.82 | 4.64 |
| Overall, I would rate the course as: | 4.85 | 4.7 |
| Overall, I would rate the teacher as: | 5.06 | 4.81 |

* Scale: 6-point Likert Scale (1: Poor/Strongly disagree <——> 6: Excellent/Strongly agree).

Performing arts education tends to have small classes, such as one-on-one or one on a few; for example, soprano coaching or dance choreography. These small classes' SFQ scores are usually high due to the close relationship between teachers and students. For data accuracy, this study excluded all small classes.

### 3.2. Scopes and Multi-Dimensional Data Model

For quality management, after collecting the requirements from the senior management, the system was designed to analyze the relationships among the SFQ four categories and the student course grade data (Figure 1).

In this study, as demonstrated in Figure 2, the analytics have been classified into four levels: "Nano-level" points to the information/data in a course, program, or department; the "Micro-level" indicates many courses in a school/faculty; the "Meso-level" includes many courses in many faculties in a specific academic year, and last, the "Macro-level" concerns many faculties and many years at the institution.

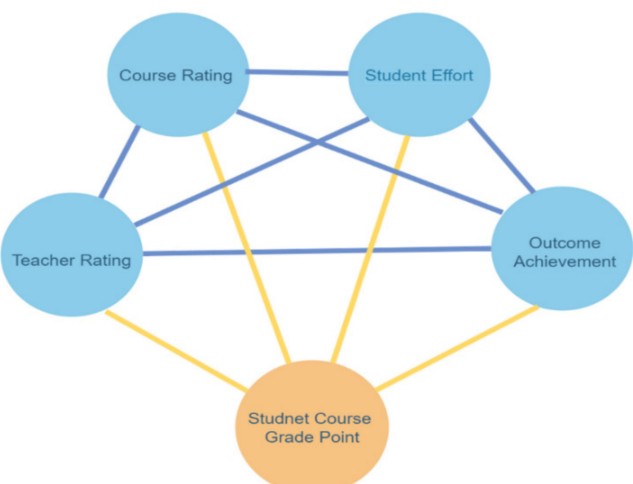

**Figure 1.** The relationship between the four SFQ categories and the student course grade to be analyzed in this study.

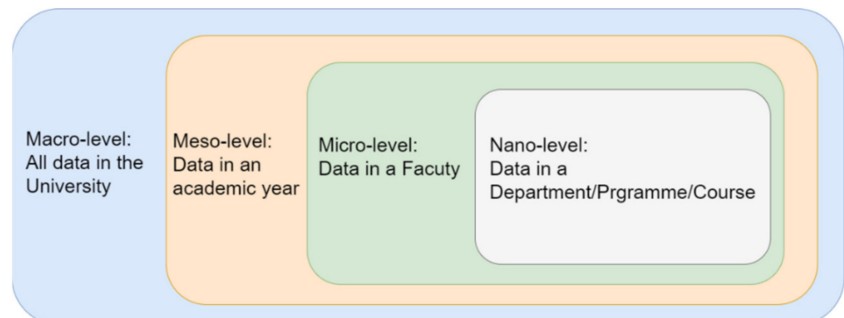

**Figure 2.** Overlapping of the analytics levels in this study.

The student Course Grade Data and Course Rating Data are included in this study in the multi-dimensional model. The dimensions are the perspectives or entities concerning which the institution keeps records. For example, the data warehouse keeps records of the Course Rating Data for the dimension time, course, and survey questions (Figure 3). These dimensions allow the ability to keep track of things. Each dimension has a table related to it, called a dimensional table, which describes the dimension further.

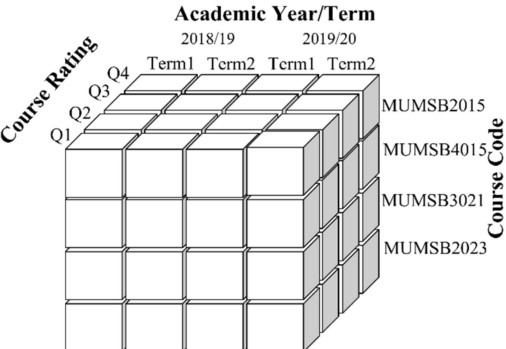

**Figure 3.** A multi-dimensional data model (3D data cube) for the course rating.

*3.3. Data Warehouse Architecture*

At semi-annual Academic Board Meetings, reports processed through this system are presented to the senior management and high-profile faculty members (deans, directors, and program leaders). The reports explain complicated data and interpretations via visual graphs with simple lines, dots, and key points. Senior management members are pleased to hear the presentation, view the graphs, and take swift and effective actions on policy implementations. However, for a broader understanding, the researchers break it down to make it relevant and applicable by others who might be interested in using a similar approach.

To achieve the multi-dimensional model in Figure 3, a data warehouse, also known as an enterprise data warehouse, a system for reporting and data analysis, is used as a core component of business intelligence [25]. The data warehouse performs sophisticated data analysis for all kinds of users, especially the senior management/decision-makers. The data warehouse supports developers to create customized and complex queries to retrieve information based on multiple data sources. It provides a systematic approach for the people who want simple technology to access the data for making decisions, especially when business users want to extract key information with fast performance from a huge amount of data in a certain format, e.g., reports, grids or charts. Figure 4 demonstrates the data flow of the survey data in the data warehouse.

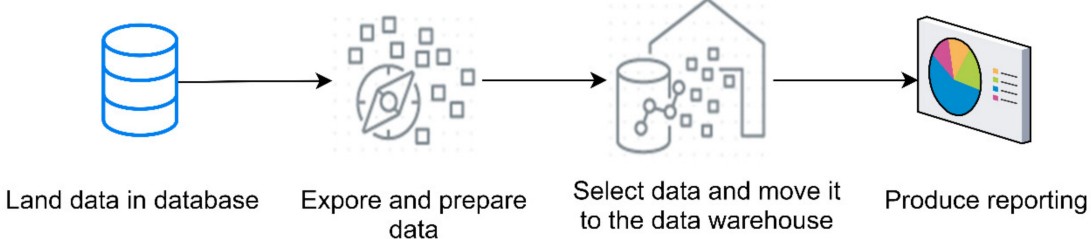

**Figure 4.** The overview of the data flow in a data warehouse.

Compared to the traditional operational database, data warehouses are optimized for analytic access patterns. Traditional operational databases are based on the following two technologies. Online transaction processing (OLTP), which is a category of data processing that is focused on transaction-oriented tasks [26]. Online analytical processing (OLAP), which is an approach to answering multi-dimensional analytical (MDA) queries swiftly in computing, which is the approach behind Business Intelligence (BI) applications. Due to the technical complexity and for a broader understanding, the researchers present it through a few graphic designs below to illustrate how it works. The difference between OLAP and OLTP is described in Figure 5.

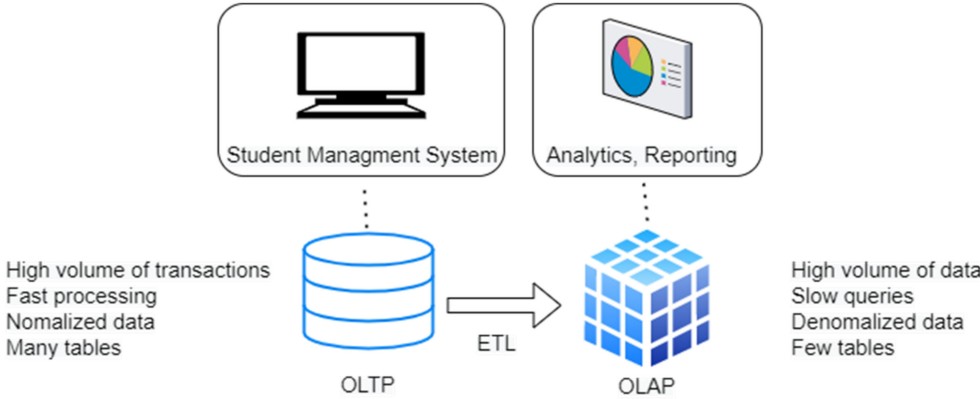

**Figure 5.** OLTP vs. OLAP.

ETL is an abbreviation of extract, transform, and load. In this study, the ETL tool (SSIS) extracts the data from different database source systems (e.g., student information, survey data). Then, it transforms these data into a staging place (format, structure) and then loads the data into the final stage in the data warehouse system every night. This is an important part not only because it generates detailed reports for semi-annual board meetings it also provides the most updated information for senior faculty leaders, such as the dean of a faculty. An example to elaborate is that the current semester is suspended due to the fifth wave of the COVID-19 pandemic. The academy has called an emergency academic board meeting. In order to make the right and evidence-based decision, this data warehouse system's ability to generate the latest report is proven to be extremely valuable. As shown in Figure 6, the raw sensor data are aggregated from the operational database by ETL and then are processed to the data warehouse, which divides them into different data marts for the reporting service.

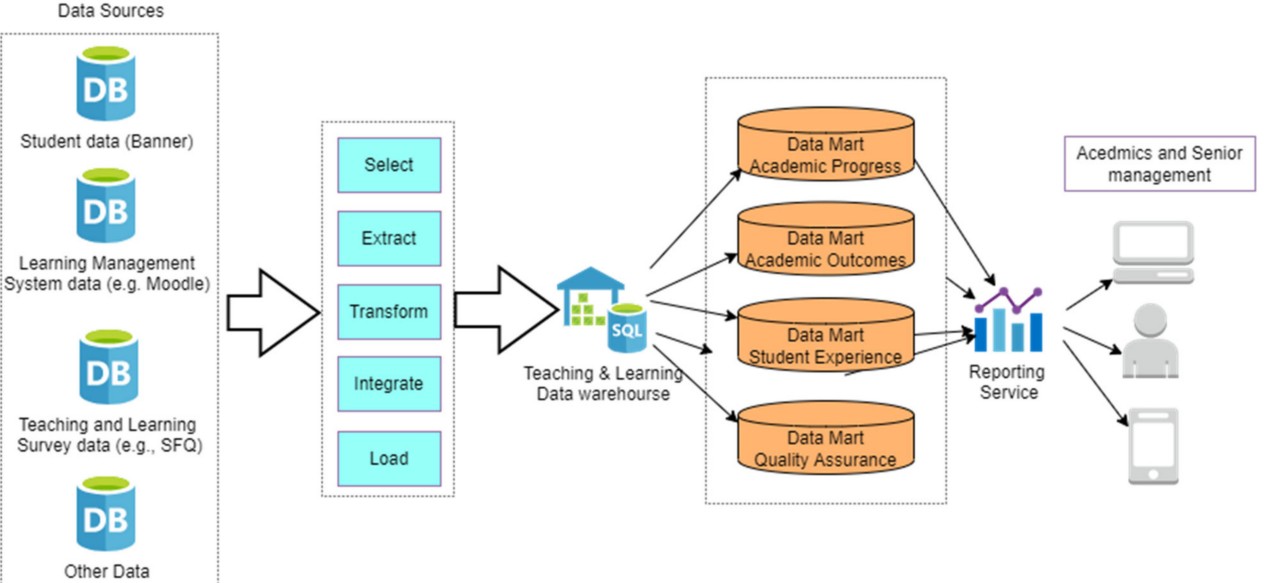

**Figure 6.** The data flow in the data warehouse in this study.

After the ETL process, a data cube is generated in the data warehouse, which is based on the multi-dimensional data model(s). Inside the data warehouse, the data allows data to be viewed and modeled in multiple dimensions and tables; the fact table contains measures of the raw data and foreign keys with related dimension tables in the data warehouse.

In this study, the development lifecycle (Figure 7) is an iterative approach based on six steps:

1. Identify the system requirements and associate value from the program leader and senior management. A broad spectrum of user requirements is collected among various academic departments/units for the desired aspects and reports. Meanwhile, the data sources (owner, availability, constraints, quality) are identified.
2. A high-performing multi-dimensional model is designed and built in the database (SQL server in this study), according to the user requirements collected from step 1.
3. Liaise with various data source owners; an IT team designs the database schema and the ETL program to extract, transform, and load the source data into the data warehouse system.
4. Based on user requirements, the data marts and CLAP cubes are built in the data warehouse, with complex measures and calculations.
5. A dynamic interactive user interface is developed in Power BI for end-users to explore the data via mobile, PC, or other applications (e.g., excel, Power Apps)
6. The developed solution is tested with end-users and deployed to the Power BI cloud.

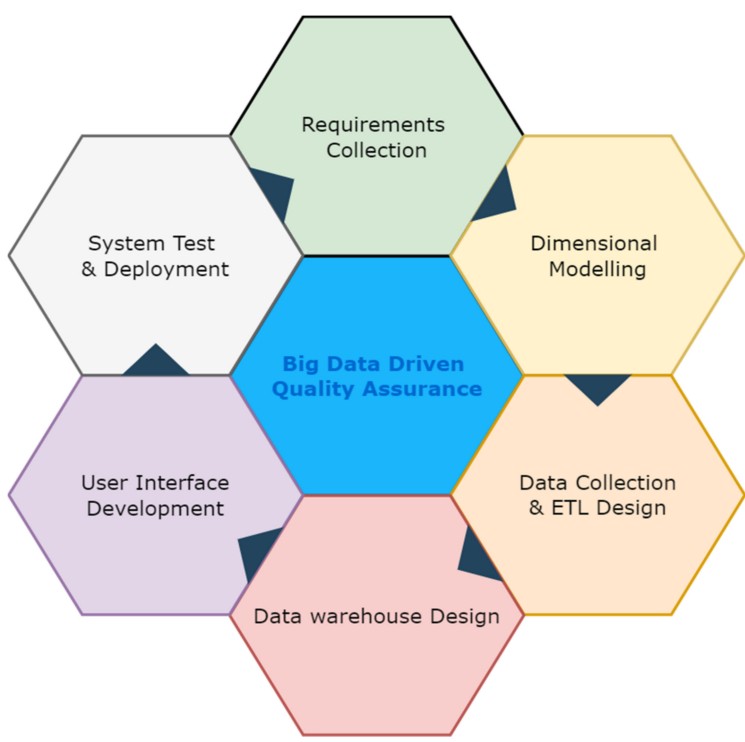

**Figure 7.** Data warehouse development lifecycle.

### *3.4. User Interface*

This data warehouse makes the collection, visualization, and interrogation of course evaluation data easily accessible for course coordinators and senior administrators. With the following user interfaces, the relevant people can make swift and necessary actions.

#### 3.4.1. Summary Statistics

A summary statistics page presents information on the number of Eligible Respondents, the number of Actual Respondents, Response Rate, and the number of Courses for teachers (Figure 8).

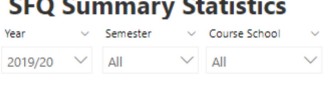

**Figure 8.** Course Evaluation Summary Statistics.

### 3.4.2. Students' Ratings on Courses

Students' ratings on courses, their feedback, Teacher Rating vs. Course Rating, and Course Rating vs. Outcome Achievement are presented in this dashboard (Figure 9). This integrated graph gives administrators quick access to all the critical data. It compiles all data in one place for deeper understanding and more robust analysis. Users can view the rating data of different academic terms and schools by selecting the slicer-"Year", "Semester", and "Course School". The slicer-"Course Section of Course Count", "Course Code", and "Course Count taught by" are used for users to filter data of the "Section number of a course", a specific course and course teachers to get deeper insight focusing on a particular course. Users can also click on the visuals below to filter the data in the report. This would lead to swift and effective action, if necessary.

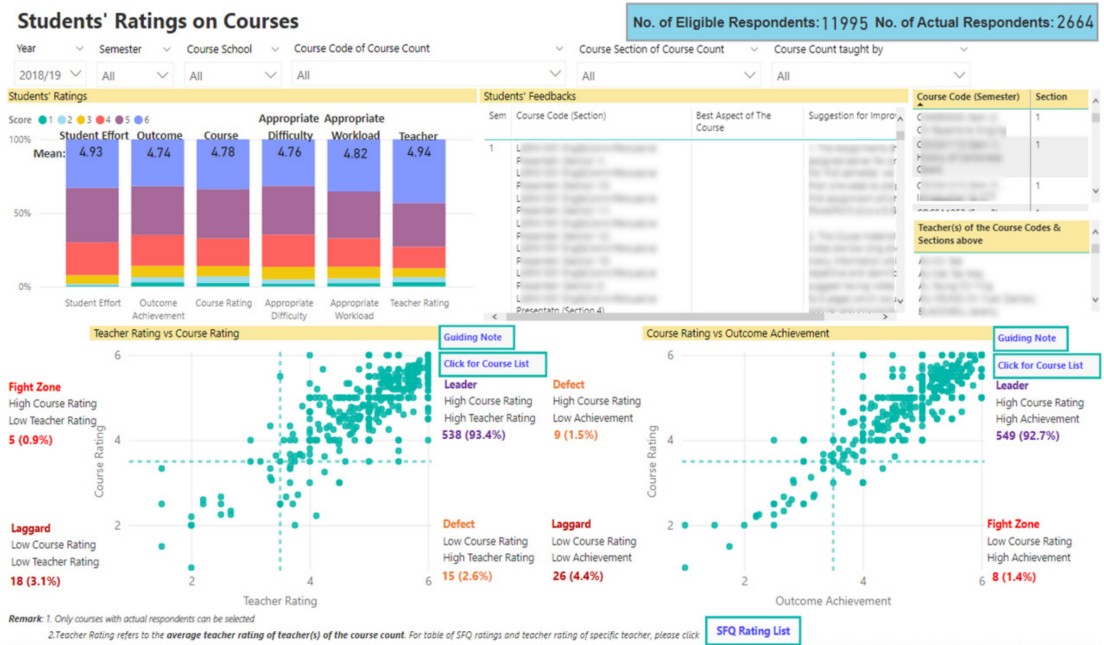

**Figure 9.** Students' Ratings of Courses; Note: (1) In the visual "Teach Rating vs. Course Rating", each data point represents the average rating given by students to a teacher for a particular course; (2) In the visual "Course Rating vs. Outcome Achievement", each data point represents the average course rating given by students to the question "Outcome Achievement" for a particular course.

### 3.4.3. Students' Ratings on Teachers

Teacher Ratings vs. feedback usefulness and students' feedback are presented in this dashboard (Figure 10). This image presents some critical information (ratings and feedback). Through color coding, summary tables, and detailed sub-score reports, administrators can quickly identify trends. They can use such data to plan differentiated interventions and enrichment activities.

### 3.4.4. SFQ Ratings vs. Average Student Grade Points

This dashboard presents Student Effort vs. Students' Grade by course and by semester; Course Rating vs. Students' Grade by course and by semester; and Teacher Rating vs. Students' Grade by course and by semester (Figure 11).

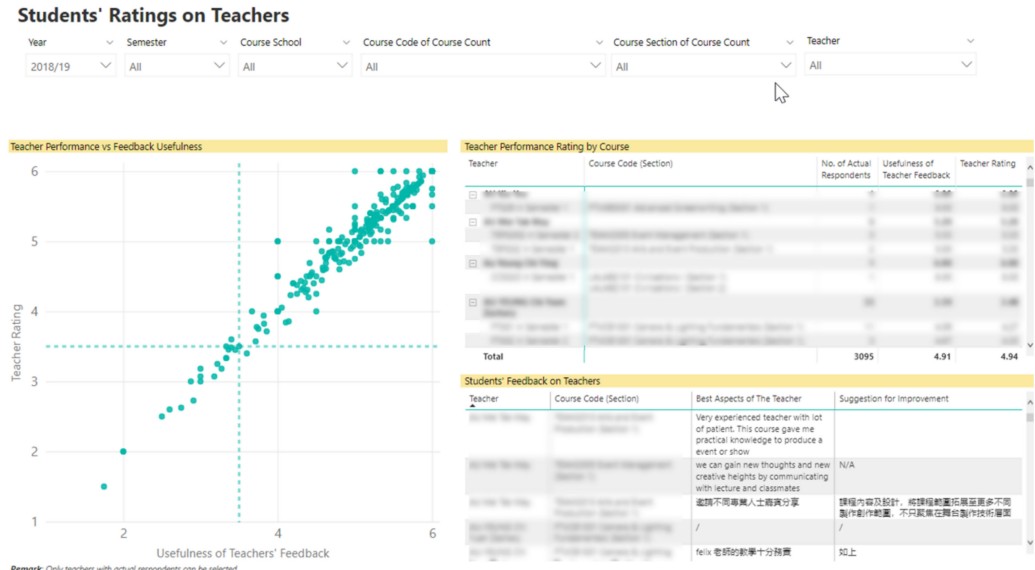

**Figure 10.** Students' Ratings of Teachers. Note: (1) In the visual "Teach Performance vs. Feedback Usefulness", each data point represents the average rating given by students to a teacher and the question "The teacher provided me with a helpful feedback" for a particular course. (2) The English explanation of the Chinese-'邀請不同的專業人士嘉賓分享' is 'Invite different professionals to share their experience'. (3) The English explanation of the Chinese-'課程內容及設計, 將課程範圍拓展至更多不同製作創作範圍, 不只聚集再舞臺製作技術層面' is 'For the course design, please extend the content scope to more diverse production creation areas, no limited in the stage production'. (4) The English explanation of the Chinese-'老師的教學十分務實' is 'The course content is very practical'. (5) The English explanation of the Chinese-'如上' is 'Same as above'.

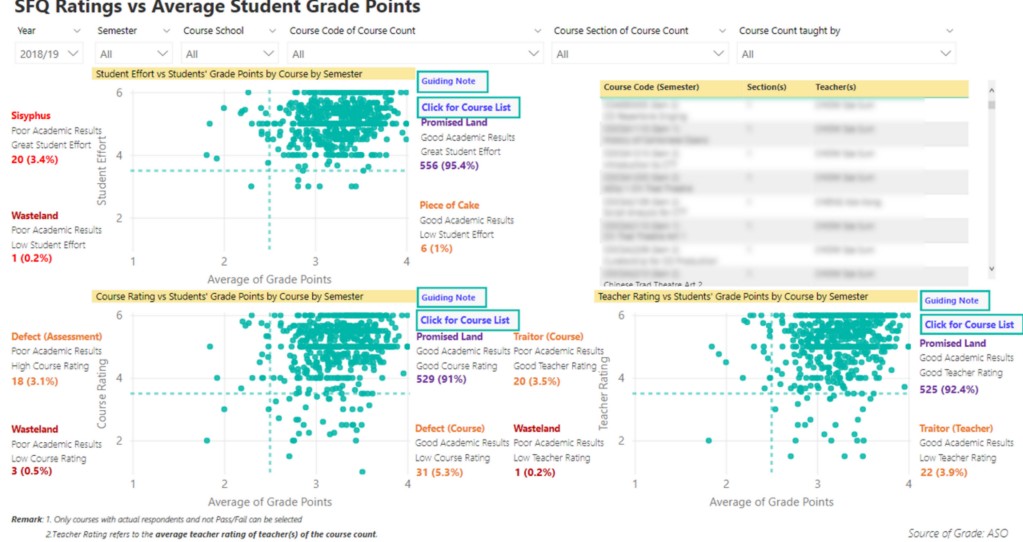

**Figure 11.** SFQ Ratings vs. Average Student Grade. Note: (1) In the visual "Student Effort vs. Students' Grade Point", each data point represents the average "Student Effort" rating given by students, and students' grade for a particular course; (2) In the visual "Course Rating vs. Students' Grade Point", each data point represents the average course rating given by students, and students' grade for a particular course; (3) In the visual "Teacher Rating vs. Students' Grade Point", each data point represents the average Teacher Rating given by students, and students' grade for a particular course.

## 4. Quality Assurance

As demonstrated in Section 3.2, the data warehouse system is designed to analyze the relationships among the SFQ four categories and the student course grade data (Figure 1). To help school and program leaders easily figure out the courses/teachers that need closer scrutiny to ascertain whether and what areas of course enhancement measures are warranted, a Novel Analytical Framework based on the data warehouse in the course evaluation data from a course management perspective, is proposed with five matrices (Course vs. Teacher Rating; Outcome Achievement vs. Course Rating; Course Average Grade Point vs. Student Effort; Course Average Grade Point vs. Course Rating; Course Average Grade Point vs. Teacher Rating) to categorize courses into various domains. Such an examination provides clear guidance on what areas of course enhancement measures are needed. The matrices scrutinize courses through multi-dimensional lenses to identify areas for improvement. Each domain contains a metaphor and differentiated enhancement strategy for addressing domain-specific course issues.

### 4.1. Course-Teacher Rating Matrix

Student ratings of teacher effectiveness are probably one of the most used sources of data for faculty evaluation [27]. From the senior management's view, the course with low Teacher Ratings and Course Ratings requires teacher training in coaching skills and careful review of the course content.

Students' overall ratings of Course and Teacher Rating are used in this matrix to categorize the courses into the following four domains: (1) "Leader", "Defect", "Fight Zone", and "Laggard". The descriptions are explained in Table 3.

**Table 3.** The definition of Course–Teacher Rating Matrix.

| | |
|---|---|
| "Leader" | Students loved the courses and the teachers. Awesome! |
| "Defect" | Low Course Rating + high Teacher Rating may indicate problems in course design and/or administrative arrangement. ***Review in these areas is warranted.*** |
| "Fight Zone" | High Course Rating + low Teacher Rating reflects poor student–teacher interaction, although students were satisfied with the course. ***Professional development intervention for those concerned teachers is automatically recommended.*** |
| "Laggard" | Low ratings in both course and teacher. A holistic review of the course design, learning support, and teaching practice is necessary. ***Relevant support and professional development suggestions are recommended.*** |

Based on this matrix, 93.3% of the courses in the 2018/19 SFQ survey are in the Leader domain, as shown in Figure 12. Data show that students are satisfied with the course, and they are happy with their subject teachers. Meanwhile, the courses in the Laggard domains are discussed in the Academic Policy and Quality Assurance Committee meeting in the university.

For instance, a diagnosis of the Diploma in Foundations (DipF) courses shows that almost all courses have their ratings on Course, Teacher, and Self-rated Outcome Achievement above 4.00 points and are in the "Leader" domains of the course matrices developed in the SFQ analysis, and only one course (highlighted) needs further improvement (Figure 13). The results suggest that the DipF students dominantly liked the courses and teachers and feel they achieved the learning outcomes. It can be concluded that the DipF program made a very successful debut in achieving satisfactory student ratings.

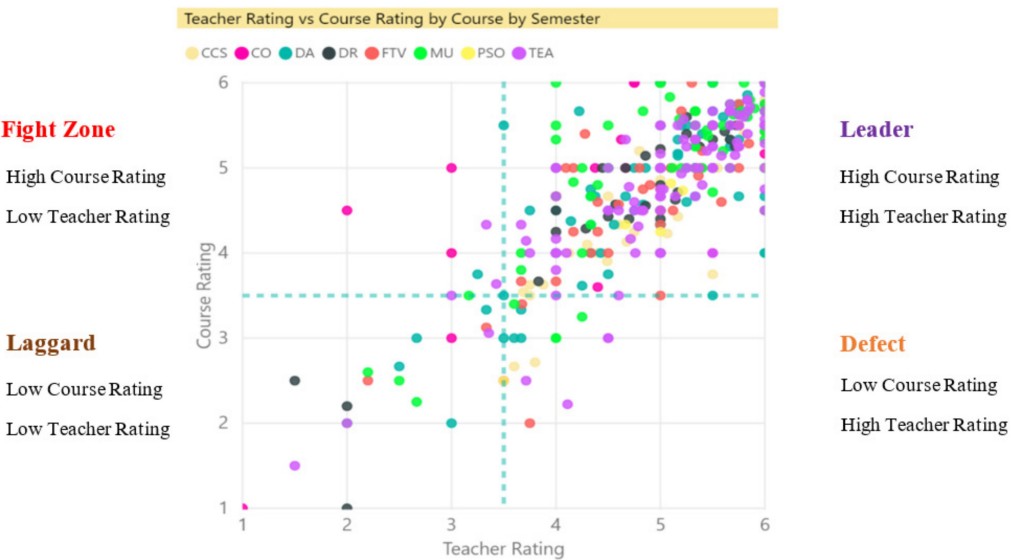

**Figure 12.** Teacher–Course Rating Matrix. Note: (1) In the visual, each data point represents the average course rate and rating given by students to a teacher for a particular course.

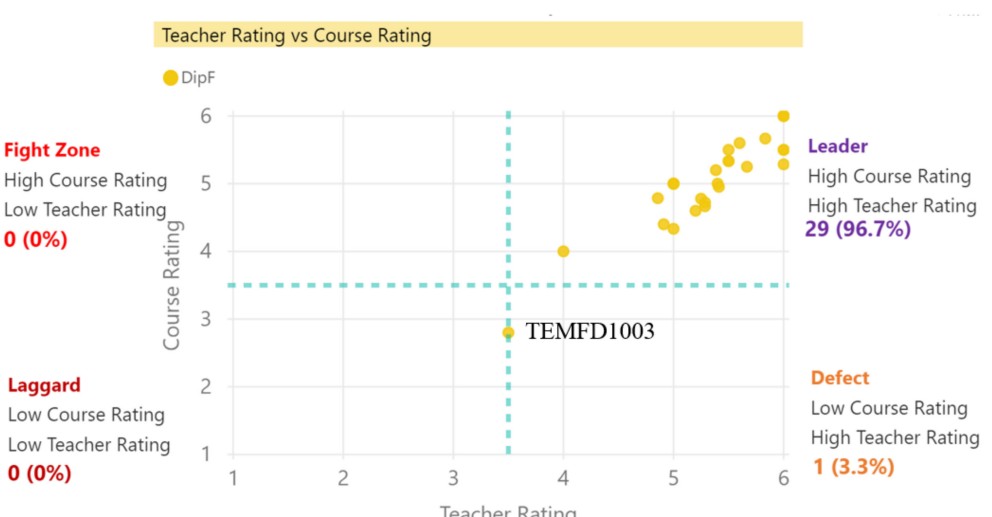

**Figure 13.** Teacher–Course Rating Matrix. Note: (1) In the visual, each data point represents the average course rate and rating given by students to a teacher for a particular course.

The data also show that teaching practices are effective when they promote students' perceived autonomy in learning through adopting active learning strategies, giving supportive feedback, and facilitating their self-directed and collaborative learning, which are referenced in studies [18,19,28]. In this context, the usefulness of teachers' feedback has a strong impact on teaching effectiveness from the students' perspectives (Figure 14). With the performance-based education approach in mind, developing teachers' skills in giving personalized, constructive feedback is extremely important for cultivating students' ability to be reflective, and it supports their artistic development in relevant fields. It is recommended to provide teacher training in communication skills so that more constructive and positive feedback is used.

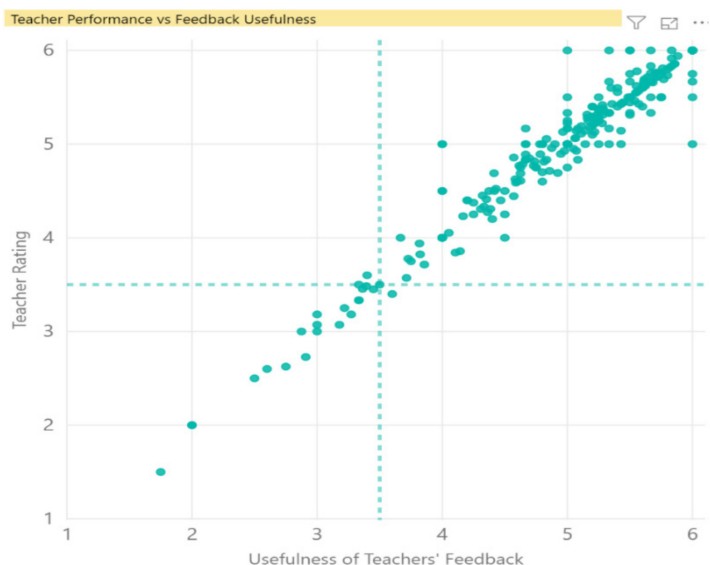

**Figure 14.** Feedback Usefulness vs. Teacher Rating. Note: (1) In the visual, each data point represents the average rating given by students to a teacher, the question "The teacher provided me with a helpful feedback" for a particular course.

### 4.2. Outcome Achievement–Course Rating Matrix

Students' overall ratings of Course Outcome Achievement and Course Rating are used in this matrix to categorize courses in to the following four domains: (1) "Leader", "Defect", "Fight Zone", and "Laggard." The descriptions are explained in Table 4.

**Table 4.** The definition of Outcome Achievement–Course Rating Matrix.

| | |
|---|---|
| "Leader" | Students felt that they had achieved their learning objectives and loved the course. |
| "Defect" | Students felt that they had achieved little but gave high Course Ratings. The courses may be too easy and not challenging enough. A review of the course's intended learning outcomes (CILOs) and constructive alignment of assessment strategies is recommended. |
| "Fight Zone" | Students felt that they had achieved a lot but hated the course. This may reflect issues in the teaching practice, workload, assessment methods, or administrative arrangement. A review of these course issues is recommended. |
| "Laggard" | Low ratings in both Course and Outcome Achievement. A holistic review of the course design, learning support, teaching practice, assessment methods, and administrative arrangement is necessary. |

In the 2018/19 academic year, 93.1% of the courses in the SFQ survey are in the "Leader" domain, a sign of satisfactory course performance from the students' perspectives (Figure 15).

### 4.3. Course Average Grade Point–Student Effort Matrix

Students' overall ratings of Self-effort and Course Average Grade Point are used in this matrix to categorize the courses into the following four domains: (1) "Promised Land", "Sisyphus", "Piece of Cake", and "Wasteland". The descriptions are explained in Table 5.

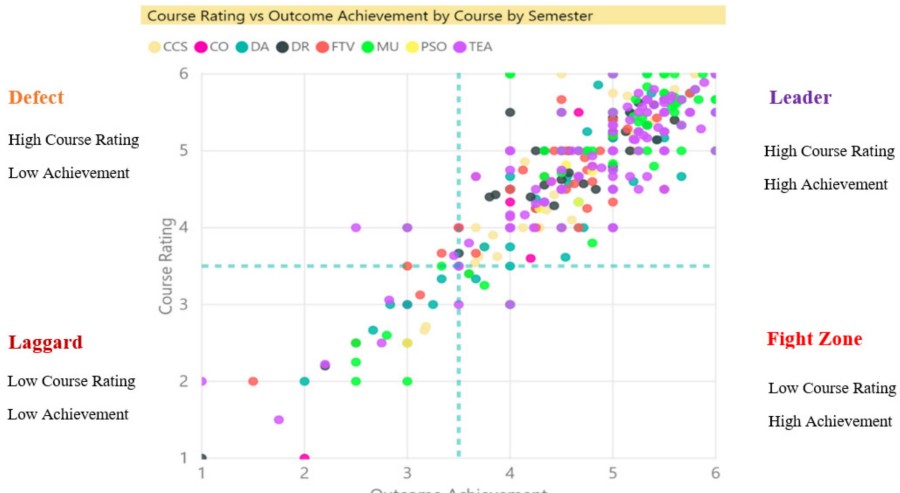

**Figure 15.** Outcome Achievement–Course Rating Matrix. Note: (1) In the visual, each data point represents the average course rating given by students to the question "Outcome Achievement" for a particular course.

**Table 5.** The definition of Course Average Grade Point–Student Effort Matrix.

| | |
|---|---|
| "Promised Land" | Students paid great effort and achieved outstanding learning outcomes. |
| "Sisyphus" | Students paid great effort but received discouraging results. It is worthy of reviewing whether the CILOs and assessments were too challenging. |
| "Piece of Cake" | Students paid little effort but got good grades. It is worthy of reviewing whether the CILOs and assessments were too easy. |
| "Wasteland" | Students were disengaged and performed poorly. A holistic review of the course design, learning support, and teaching practice is necessary. |

When students study in a social, friendly atmosphere, they feel safe to learn from problem-solving and also from each other. Teaching practices that target high-order cognitive skills, setting challenging yet manageable learning goals are found to stimulate student engagement. In some cases, that results in a "flow" experience in which the students show full concentration, interest, and joy in learning [1,3,29]; as such, developing CILOs and appropriate assessment strategies based on students' stages of artistic development is vital for effective teaching and meaningful learning. This is especially evident in performing arts contexts.

Figure 16 shows the matrix using the average grade point of a course as an "objective" measure of students' academic achievement against their perceived effort. The grade point of 2.5, i.e., the midpoint of B- and C+, is used as the threshold. In the 2018/19 academic year, 95.5% of the courses are situated in the "Promised Land" domain, suggesting that the majority of the courses have set appropriate difficulty levels for students.

*4.4. Course Average Grade Point–Course Rating Matrix*

Students' overall ratings of the Course and the Course Average Grade Point are used in this matrix to categorize the courses into the following four domains: (1) "Promised Land", "Defect (Course)", "Defect (assessment)", and "Wasteland". The descriptions are explained in Table 6.

In this matrix, "Defect" domains refer to the courses with high course ratings but received lower than average grades and vice versa. The former case may warrant a review

of assessment strategies, and the latter one may need to pay attention to issues related to course arrangement and teaching strategies. In the 2018/19 academic year, 91.7% of courses were located in the "Promised Land" domain, a strong indication that the majority of students have had a good learning experience (Figure 17).

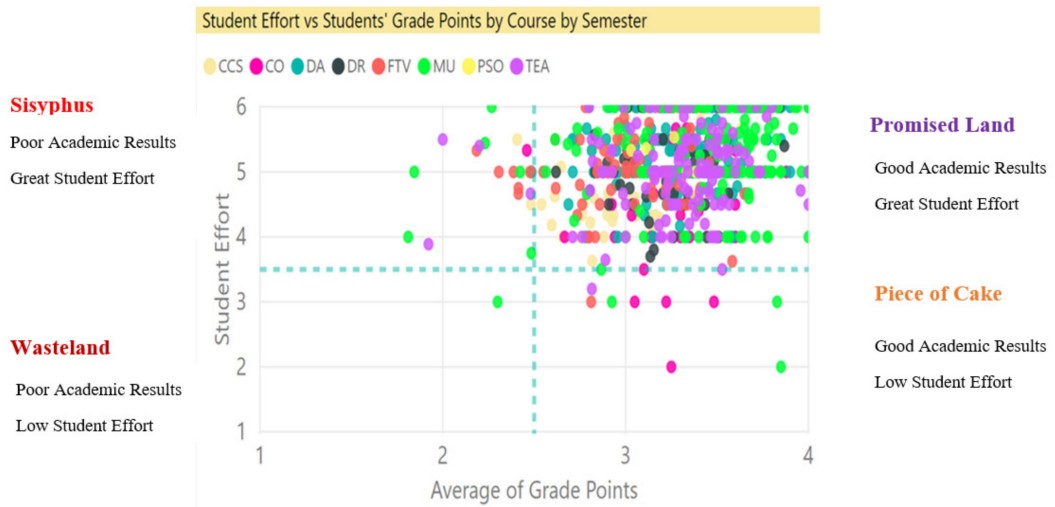

**Figure 16.** Course Grade Point–Student Effort Matrix. Note: (1) In the visual, each data point represents the average "Student Effort" rating given by students, and students' grade for a particular course.

**Table 6.** The definition of the Course Average Grade Point–Course Rating Matrix.

| | |
|---|---|
| "Promised Land" | Students loved the courses and achieved outstanding learning outcome. |
| "Defect (Course)" | Low Course Rating and outstanding results. It is worthy of reviewing whether the CILOs and assessments were too challenging. |
| "Defect (assessment)" | High Course Rating but got discouraging grades. It is worthy of reviewing whether the CILOs and assessments were too easy. |
| "Wasteland" | Low ratings in course and students performed poorly. A holistic review of the course design, learning support, and teaching practice is necessary. |

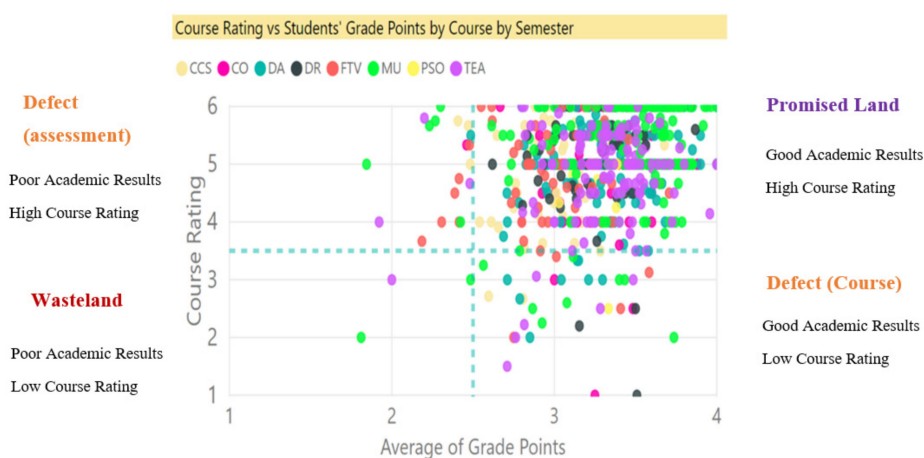

**Figure 17.** Grade Points–Course Rating Matrix. Note: (1) In the visual, each data point represents the average course rating given by students, and students' grade for a particular course.

### 4.5. Course Average Grade Point–Teacher Rating Matrix

Students' overall ratings of the Course and the Course Average Grade Point are used in this matrix to categorize the courses into the following four domains: (1) "Promised Land", "Defect (Course)", "Defect (Teacher)", and "Wasteland." The descriptions are explained in Table 7.

**Table 7.** The definition of Course Average Grade Point–Teacher Rating Matrix.

| | |
|---|---|
| "Promised Land" | Students loved the teachers and achieved outstanding learning outcome. |
| "Defect (Course)" | High Teacher Rating but got discouraging results. It is worthy of reviewing whether the CILOs and assessments were too challenging. |
| "Defect (Teacher)" | Low Teacher Rating but got good grades. It is worthy of reviewing whether the CILOs and assessments were pitched too easy. |
| "Wasteland" | Low ratings for the course and students performed poorly. A holistic review of the course design, learning support, and teaching practice is necessary. |

In the "Defect (Course)" domain (Figure 18), students enjoyed taking the course and liked the teacher, but they performed poorly. In such a case, a review of the course curriculum design and assessment strategies is recommended.

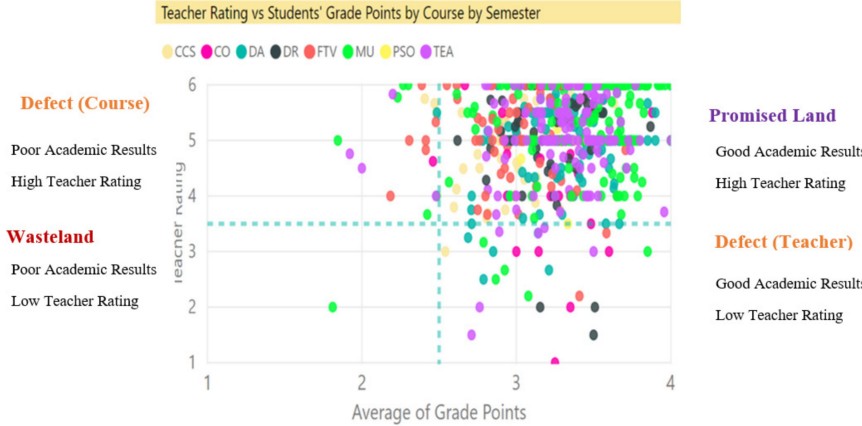

**Figure 18.** Average Grade Points–Teacher Rating Matrix. Note: (1) In the visual, each data point represents the average Teacher Rating given by students, and students' grade for a particular course.

Courses in the "Defect (Teacher)" domain may be attributed to students' negative response to demanding teachers [30]. Studies show that student engagement and performance are fostered by the teachers' approach to teaching and learning environments that satisfy the students' needs for social relatedness, autonomy, and competence [1,3,15]. Professional development intervention to enhance teachers' skills in supportive teaching is recommended. Specifically, peer mentoring, as a form of teacher support, is a proven strategy for improving teaching effectiveness [31]. Peer mentoring not only helps new faculty members to learn effective teaching strategies from seasoned educators but also offers psychological support. This has a long-term benefit as peer mentoring helps new staff see mentoring as a career-enhancing activity [32,33] rather than a task that they have to complete.

### 5. Conclusions and Discussion

This study applies multi-dimension lenses to analyze the large volume of data in the course student evaluation area, a key aspect of the teaching and learning quality assurance

for higher education. Based on the data warehouse system, a novel analytical framework is proposed for categorizing the courses into various domains (Student Effort, Outcome Achievement, Course Rating, and Teacher Rating) to address domain-specific course issues, which helps senior administrators and program leaders quickly identify courses that need closer scrutiny. This streamlined system ascertains whether and what areas of course enhancement measures are warranted.

This novel analytical framework provides efficient assessment and leads to effective solution-based actions. Such a measure enables policymakers and senior management to see the big picture and gain insight from a large volume of the student course and teacher evaluation data. It analyzes a huge amount of data, including teaching and learning, activities, assessment, performance quality, among other domains. For instance, the use of this framework generates recommendations to pair up those teachers in the "Promised Land" with those struggling in the "Wasteland" and "Traitor" domains in a peer mentoring program. Furthermore, the scales of the four domains in the five matrices of the framework need to be customized among different course types (required course/elective course and online course/face-2-face course).

Overall, this study suggests that more performing arts institutions should utilize this novel analytical framework as a diagnostic tool to provide efficient and effective oversight of their course portfolio, to identify those courses worthy of special attention, and take enhancement measures accordingly. In addition, policymakers and senior management can make use of the framework to quickly adjust education policies, develop concrete, multi-faceted feedback, and professional development strategies. Based on this framework, they can also offer constructive advice to different academic programs and associate staff during program and teacher performance reviews.

Through color coding, summary tables, and detailed sub0score reports, this framework creates an efficient workflow to identify areas of concerns makes solution-based enhancement strategies, provides evidence-based advice, monitors programs/staff effectiveness, and promotes teaching excellence. In addition, this framework could be easily implemented in other disciplines and subject areas for similar benefits in other tertiary institutions.

**Author Contributions:** Conceptualization: Q.L. and Z.L.; methodology: Q.L., Z.L. and J.H.; software: Q.L. and H.M.; writing—original draft preparation: Q.L. and Z.L.; writing—review and editing: Z.L., J.H. and H.M. All authors have read and agreed to the published version of the manuscript.

**Funding:** This research received no external funding.

**Institutional Review Board Statement:** Not applicable.

**Informed Consent Statement:** Not applicable.

**Data Availability Statement:** Not applicable.

**Conflicts of Interest:** The authors declare no conflict of interest.

## Appendix A

**Student Feedback Questionnaire**
**學生意見調查**
**2018/19 Semester 2 / Summer Term**

This Student Feedback Questionnaire (SFQ) is a way for us to collect feedback on how you felt about aspects of the course. Your feedback is extremely important to the Academy for enhancement quality and student learning experience.

本學生意見調查問卷旨在收集你對本科目的意見。你的意見對日後提升本科目質素與改善學生的學習經驗非常重要。

**Instructions**
1. You have 10-15 minutes to complete this questionnaire.
2. This survey is completely anonymous so you do not need to write your name or Student ID. You are, therefore, encouraged to provide honest feedback. Please note, however, that malicious, inappropriate comments or illegible words will be ignored.
3. Please complete the questionnaire by yourself. Do not discuss the questions with anyone.
4. Please use **Black / Blue ball pen or Pencil ONLY** to fill up the selected oval completely. A rating of "6" is the most positive while a rating of "1" is the most negative.
   Sample: ○ ○ ○ ● ○ ○
5. For corrections, please use erasers or correction fluid.
6. Please complete Part F of this questionnaire in Chinese / English only.

**填寫須知**
1. 請用 10 至 15 分鐘填寫此問卷。
2. 你無須填寫姓名或學生編號，請誠實作答所有問題。唯惡意、不恰當或無法辨認字體的意見將不獲處理。
3. 請自行填寫此問卷，不要與別人討論。
4. 請**使用藍 / 黑色原子筆或鉛筆**清晰地填滿代表你的意見的數字。請注意，6 分代表最滿意，1 分代表最不滿意。
   例：○ ○ ○ ● ○ ○
5. 如有更改，請使用擦膠或塗改液。
6. 請用中文/英文填寫己部。

**Part A: General information　甲部：一般資料**

**My Programme of Study　我所就讀課程**

**Certificate / Diploma / Advanced Diploma　證書 / 文憑 / 深造文憑 / 高等文憑**
- ○ Certificate in Theatre and Entertainment Arts (Fast Track Vocational) 舞台及製作藝術(精研職業訓練)證書
- ○ Diploma in Cantonese Opera 粵劇文憑
- ○ Diploma in Dance 舞蹈文憑
- ○ Diploma in Music 音樂文憑
- ○ Advanced Diploma in Cantonese Opera 粵劇深造文憑 / 粵劇高等文憑
- ○ Advanced Diploma in Dance 舞蹈深造文憑 / 舞蹈高等文憑
- ○ Advanced Diploma in Music 音樂深造文憑 / 音樂高等文憑

**Bachelor's Degree　學士**
- ○ BFA in Chinese Opera 戲曲藝術學士（榮譽）學位
- ○ BFA in Dance 舞蹈藝術學士（榮譽）學位
- ○ BFA in Drama 戲劇藝術學士（榮譽）學位
- ○ BFA in Film and Television 電影電視藝術學士（榮譽）學位
- ○ BFA in Theatre and Entertainment Arts 舞台及製作藝術學士（榮譽）學位
- ○ BMus 音樂學士（榮譽）學位

**Master's Degree　碩士**
- ○ MFA in Cinema Production 電影製作藝術碩士
- ○ MFA in Dance 舞蹈藝術碩士
- ○ MFA in Drama 戲劇藝術碩士
- ○ MFA in Theatre and Entertainment Arts 舞台及製作藝術碩士學位
- ○ MMus 音樂碩士

**Exchange/Visiting/Others　交換/訪問/其他**
- ○ Exchange Student 交換學生
- ○ Visiting Student 訪問學生
- ○ Others, please specify 其他，請註明：
  _______________________

| | |
|---|---|
| Course title 科目名稱 | |
| Course Code 科目編號 | CRN code CRN 編號　　　　　Section no. 節數 |
| Type of course 科目類別 | ○ Required 必修　　　　○ Elective 選修　　　　○ Major Study 主修科目　　○ Contextual Study 輔助學科　　○ Languages 語文　　○ Liberal Arts Studies 人文學科 |

■

SFQ (AM1)

**Part B: My Effort 乙部：個人努力**

| | Poor 劣 | | | | | Excellent 優 |
|---|---|---|---|---|---|---|
| | 1 | 2 | 3 | 4 | 5 | 6 |

**B1.** **Overall, I would rate my effort on this course as…**
　　總括而言，我對本科目所付出的努力是……
　　　☐　☐　☐　☐　☐　☐

B2.　On average, how many hours per week did you spend on this course (not including class/contact hours with a teacher)?
　　平均每星期投放於本科目的時間 (不包括上課時間)

**hours** 小時

_______________

**Part C: My Achievement in Learning Outcomes 丙部：學習成果**

Strongly disagree 非常不同意 　　　　　Strongly agree 非常同意

C1.　I understood what I was expected to achieve in the course.
　　我明白本科目期望我須達成的目標。
　　☐　☐　☐　☐　☐　☐

C2.　The assignments effectively tested what I have achieved in the course.
　　本科目的功課能有效地評核我在本科所達成的目標。
　　☐　☐　☐　☐　☐　☐

**C3.** **Overall, by the end of the course I had acquired the expected knowledge/ skills.**
　　總括而言，我能夠獲取到預期的知識和技術。
　　☐　☐　☐　☐　☐　☐

**Part D: The Course　丁部: 本科目**

Strongly disagree 非常不同意 　　　　　Strongly agree 非常同意

D1.　I found the course engaging.　我能投入學習本科目。
　　☐　☐　☐　☐　☐　☐

D2.　The level of difficulty was appropriate.　本科目難度適中。
　　☐　☐　☐　☐　☐　☐

D3.　The workload was appropriate.　工作量適當。
　　☐　☐　☐　☐　☐　☐

D4.　The course was effective in helping me acquire the expected knowledge/skills.
　　本科目能有效地能幫助我得到預期的知識/技術。
　　☐　☐　☐　☐　☐　☐

Poor 劣 　　　　　Excellent 優

**D5.** **Overall, I would rate the course as…** 總括而言，我對本科目的評價是……
　　☐　☐　☐　☐　☐　☐

**Part E: The Teacher　戊部: 導師**

**Teacher 1 (teacher's full name): _______________________________**

Strongly disagree 非常不同意 　　　　　Strongly agree 非常同意

E1.　The teacher was effective in helping me acquire the expected knowledge/skills.
　　導師能有效地能幫助我得到預期的知識/技術。
　　☐　☐　☐　☐　☐　☐

E2.　The teacher provided me with helpful feedback.　導師能提供有用的反饋
　　☐　☐　☐　☐　☐　☐

Poor 劣 　　　　　Excellent 優

**E3.** **Overall, I would rate the teacher as…** 總括而言，我對導師的評價是……
　　☐　☐　☐　☐　☐　☐

**Teacher 2 (teacher's full name): _______________________________**

Strongly disagree 非常不同意 　　　　　Strongly agree 非常同意

E1.　The teacher was effective in helping me acquire the expected knowledge/skills.
　　導師能有效地能幫助我得到預期的知識/技術。
　　☐　☐　☐　☐　☐　☐

E2.　The teacher provided me with helpful feedback.　導師能提供有用的反饋
　　☐　☐　☐　☐　☐　☐

Poor 劣 　　　　　Excellent 優

**E3.** **Overall, I would rate the teacher as…** 總括而言，我對導師的評價是……
　　☐　☐　☐　☐　☐　☐

■

SFQ (AM1)

**Part E: The Teacher** 戊部: 導師

**Teacher 3 (teacher's full name): ______________________________**

| | Strongly disagree 非常不同意 | | | | Strongly agree 非常同意 |
|---|---|---|---|---|---|

E1.   The teacher was effective in helping me acquire the expected knowledge/skills.
導師能有效地能幫助我得到預期的知識/技術。

E2.   The teacher provided me with helpful feedback.   導師能提供有用的反饋

| | Poor 劣 | | | | Excellent 優 |
|---|---|---|---|---|---|

**E3.   Overall, I would rate the teacher as...** 總括而言，我對導師的評價是……

**Teacher 4 (teacher's full name): ______________________________**

| | Strongly disagree 非常不同意 | | | | Strongly agree 非常同意 |
|---|---|---|---|---|---|

E1.   The teacher was effective in helping me acquire the expected knowledge/skills.
導師能有效地能幫助我得到預期的知識/技術。

E2.   The teacher provided me with helpful feedback.   導師能提供有用的反饋

| | Poor 劣 | | | | Excellent 優 |
|---|---|---|---|---|---|

**E3.   Overall, I would rate the teacher as...** 總括而言，我對導師的評價是……

**Teacher 5 (teacher's full name): ______________________________**

| | Strongly disagree 非常不同意 | | | | Strongly agree 非常同意 |
|---|---|---|---|---|---|

E1.   The teacher was effective in helping me acquire the expected knowledge/skills.
導師能有效地能幫助我得到預期的知識/技術。

E2.   The teacher provided me with helpful feedback.   導師能提供有用的反饋

| | Poor 劣 | | | | Excellent 優 |
|---|---|---|---|---|---|

**E3.   Overall, I would rate the teacher as...** 總括而言，我對導師的評價是……

**Teacher 6 (teacher's full name): ______________________________**

| | Strongly disagree 非常不同意 | | | | Strongly agree 非常同意 |
|---|---|---|---|---|---|

E1.   The teacher was effective in helping me acquire the expected knowledge/skills.
導師能有效地能幫助我得到預期的知識/技術。

E2.   The teacher provided me with helpful feedback.   導師能提供有用的反饋

| | Poor 劣 | | | | Excellent 優 |
|---|---|---|---|---|---|

**E3.   Overall, I would rate the teacher as...** 總括而言，我對導師的評價是……

**Teacher 7 (teacher's full name): ______________________________**

| | Strongly disagree 非常不同意 | | | | Strongly agree 非常同意 |
|---|---|---|---|---|---|

E1.   The teacher was effective in helping me acquire the expected knowledge/skills.
導師能有效地能幫助我得到預期的知識/技術。

E2.   The teacher provided me with helpful feedback.   導師能提供有用的反饋

| | Poor 劣 | | | | Excellent 優 |
|---|---|---|---|---|---|

**E3.   Overall, I would rate the teacher as...** 總括而言，我對導師的評價是……

■

**Part F:   Qualitative Feedback   己部：反饋**

| What were the best aspect(s) about the course?<br>本科目最好的地方是甚麼？ | What aspect(s) about the course could be improved?<br>本科目有甚麼地方可以改進？ |
|---|---|
| What were the best aspect(s) about the teaching?<br>本教學最好的地方是甚麼？<br><br>Teacher 1 (teacher's full name):________________________<br><br><br>Teacher 2 (teacher's full name):________________________<br><br><br>Teacher 3 (teacher's full name):________________________<br><br><br>Teacher 4 (teacher's full name):________________________<br><br><br>Teacher 5 (teacher's full name):________________________<br><br><br>Teacher 6 (teacher's full name):________________________<br><br><br>Teacher 7 (teacher's full name):________________________ | What aspect(s) about the teaching could be improved?<br>本教學有甚麼地方可以改進？<br><br>Teacher 1 (teacher's full name):________________________<br><br><br>Teacher 2 (teacher's full name):________________________<br><br><br>Teacher 3 (teacher's full name):________________________<br><br><br>Teacher 4 (teacher's full name):________________________<br><br><br>Teacher 5 (teacher's full name):________________________<br><br><br>Teacher 6 (teacher's full name):________________________<br><br><br>Teacher 7 (teacher's full name):________________________ |
| Other feedback   其他意見 | |

Thank you for completing the questionnaire.多謝完成此問卷調查。

■

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
