# Peer review of "Quality Assurance for Performing Arts Education: A Multi-Dimensional Analysis Approach"

_applsci, doi:10.3390/app12104813_

Round 1
Reviewer 1 Report
The contribution is supported by an extensive bibliography.
The content is clear and innovative, and allows methodological proposals of interest in the field of Artistic Education, due to the optimal results that they guarantee.
The contents are supported by tables, graphs and figures very well planned.
Author Response
Dear Reviewer,
This makes me feel like flying. Thank you very much for your kind words.
We hope that this paper can help scholars and readers. Our goal is to inspire others seeking similar approach to convey complex educational issues and problems to those in power.
Again, highly appreciate your constructive comments and feedback.
Reviewer 2 Report
The authors have presented an information system that visualizes feedback provided by learners of an arts university via standard Likert scale responses.
There are some interesting points in the paper, which might merit publication if re-organized and focused on what is novel. In its current form, I feel it lacks focus and direction.
Specifically:
The abstract talks about the covid-19 pandemic. This is clearly irrelevant as the study is based on data gathered in 2018-2019.
A large part of the paper is devoted to the presentation of the software architecture. The presented architecture has no influence on the actual research (the visualization presented as a result could have easily be achieved by excel, google sheets or any other spreadsheet application without even having a database or a specialized software system). None of the presented features is relevant to the stated goals of the research; perhaps the presentation of this architecture would be interesting as a standalone conference publication, but combined with the data analytics presented in this work only changes the subject and adds confusion.
"Big data" is used very loosely. No specific big data techniques are used in any part of the analysis. It seems that the authors use the term to refer to the number of student responses they have considered, but that is not enough to justify the use of the term.
The authors refer to the use of a multidimensional "model"; I am not sure what this refers to. I understand that they combine more than one dimensions in their visualizations, but I don't see where they have defined a model for the data.
The definition and labeling of the different areas in the visualizations is, to me, the most interesting part of the paper. Unfortunately, this is treated superficially. It would be interesting if the work examined the different areas in different detail, examining the differences between the cases that fall in different areas and - particularly - explaining how this information can be used by the higher management, as is the stated goal of the paper. Such an analysis would make the work, in my view, very interesting and worthy of journal publication. It would also bring it closer to justifying the use of the term big data.
The different labels in each visualization seem interesting in theory, but are proven less meaningful when seen in the real life examples as almost all cases fall in only one of the four labels. A comment on this by the authors would be useful.
Although real life data from many courses are used and visualized, no example is provided of the kind of conclusions that can be derived and the way these conclusions can be put to practical use. Without such an example, we are missing the proof of concept, i.e. an indication that this approach has any actual practical meaning. I have no doubt that such examples could be found, but this is not something that should be left to the reader's imagination. It is connected to the motivation for the paper and needs to be clearly stated.
As an overall conclusion, I find the examination of such rich data, its visualization via the combination of different parameters and the labeling of visualization areas as very interesting ideas that form an excellent basis for a paper. On the other hand, I find the focus on the software architecture, the lack of depth in the discussion and the loose use of scientific terms as weaker points of the work. Should the latter be addressed, I believe there is the potential for an excellent article.
Reviewer 3 Report
While the work presented is an important first step towards the goals set by the authors, I argue that, in its current state, it does not provide enough evidence of its suitability and value to address the intended problems. Without being deeply thorough over some of the aspects missing or that can be improved, I would say that:
- One important aspect missing concerns the lack of clear requirements for the system. Beyond the overall purpose of assessing teaching and learning, what are concrete questions that senior management wants to answer? Without these requirements and clear goals, it is impossible to effectively assess if the system is appropriate.
- The end of section 2 and the conclusions claim that the proposed framework results in "an easy and effective visualization for ...." and "... framework provides efficient assessment, and leads to effective solution-based actions.". To the best of what I could grasp no evidence is provided in this regard. To be able to claim that is easy and effective, it has to be put to test by users. And what are "solution-based actions"?
- From my point of view, while a development lifecycle is presented in Figure 5 as having been adopted for this work, the evidence provided in the manuscript is far from validating this claim.
- The chosen visualizations, adopting a matrix representation, are OK to explore pairs of variables, but what if, for instance, the user wants to have a deeper insight focusing on a particular course? Wouldn't multidimensional (beyond 2) representations be more appropriate? Isn't this kind of analysis important? As it is, it seems there is never a really hollistic view over the phenomenon. Is this not relevant? Once again, having no clear requirements makes it harder to understand up to where it should go.
- On one hand, the system seems based on data that is available (updated) once every semester, the time when the presented visualizations are possible. But, at one point, it is mentioned the data warehouse is refreshed quite often (line 203, "every night". Why?
- All figures need self-explanatory captions of what is being shown, explaining what exactly are the data points, what their colors mean, etc. For instance "each data point represents the average rate given by students to a teacher, in a particular course."
- Perhaps having an explanation about what is being explored for each of the analysis (e.g., course-teacher - subsections of section 4) could improve clarity. The rationale is somehow in the different tables, but having it in the text may help. This would emphasize the question(s) being addressed.
- One important aspect that seems to be missing is that, in my humble opinion, the authors do not really establish the value of what they have compared with what was previously available. What can I take from this system that was impossible for me to have without it? For instance, let us consider I have the data from the questionnaires: if a teacher has a rating below 3, I will take measures; if a student has a score below 3, I take measures. And I can do this without the visualizations. So, what is the added value of this system that makes it an important addition?
- Regarding the different pairs of variables covered by the visualizations, these are chosen with a concrete purpose and the classifications for the different positions of the matrix reflect how each data point is positioned in that domain. So, in my humble perspective, these seem like the pillars of the authors' Quality Assurance framework? Maybe explaining these 4 pillars at the beginning of section 4 - succintly highlighting their key contribution-, with a diagram could help strengthen the conceptual value of the work?
- It might be a positive aspect for understanding the value of what is proposed if the authors presented a walkthrough of what would be a typical analysis, using the different data and visualizations to reach a concrete conclusion about a situation.
- I understand that having an insight over the current state of things is important, but having a context of previous performance seems relevant to assess the seriousness of a particular case or evolutions. To the best of what I could grasp, this is not possible. Why?
- As it is, the work is presented as a finished system - made somehow evident by no mention to possible future evolutions/improvements. I would argue that several limitations may need to be identified and a range of novel methods may be explored for enhanced data exploration.
Round 2
Reviewer 2 Report
In my review of the original version of this article I identified a high potential that was unfortunately not highlighted in the text. My honest assessment was that the authors could use this work to produce an article that merits journal publication, but that many weeks of work would be required.
I find myself happily surprised that the authors have, in such a short time, adequately addressed all my concerns.